# Extracorporeal Shock Wave Therapy Improves In Vitro Formation of Multilayered Epithelium of Oral Mucosa Equivalents

**DOI:** 10.3390/biomedicines10030700

**Published:** 2022-03-18

**Authors:** Katharina Peters, Nadine Wiesmann, Diana Heimes, Roxana Schwab, Peer W. Kämmerer, Bilal Al-Nawas, Ronald E. Unger, Annette Hasenburg, Walburgis Brenner

**Affiliations:** 1Department of Obstetrics and Gynecology, University Medical Center of the Johannes Gutenberg University Mainz, 55131 Mainz, Germany; kpeter01@students.uni-mainz.de (K.P.); roxana.schwab@unimedizin-mainz.de (R.S.); annette.hasenburg@unimedizin-mainz.de (A.H.); 2Department of Oral and Maxillofacial and Plastic Surgery, University Medical Center of the Johannes Gutenberg University Mainz, 55131 Mainz, Germany; nwiesman@uni-mainz.de (N.W.); diana.heimes@unimedizin-mainz.de (D.H.); peer.kaemmerer@unimedizin-mainz.de (P.W.K.); al-nawas@uni-mainz.de (B.A.-N.); 3Department of Otorhinolaryngology, Head and Neck Surgery, University Medical Center of the Johannes Gutenberg University Mainz, 55131 Mainz, Germany; 4Institute of Pathology, University Medical Center of the Johannes Gutenberg University Mainz, 55131 Mainz, Germany; runger@uni-mainz.de

**Keywords:** extracorporeal shock wave therapy, oral mucosa equivalent, tissue engineering, multilayered epithelium, basement membrane, vascularization

## Abstract

Oral mucosa is used in various surgical fields as a graft for the reconstruction of tissue defects. Tissue engineering of oral mucosa equivalents using autologous cells represents a suitable less burdensome alternative. The survival of the multilayered epithelium is essential for the functionality of the tissues in vivo. To ensure its functionality after transplantation, mucosa equivalents in vitro were subjected to extracorporeal shock wave therapy (ESWT) to determine whether this treatment stimulated the formation and differentiation of the epithelium. Mucosa equivalents treated with ESWT were examined for cellular metabolic activity using AlamarBlue^TM^ assay. The formation of vascular structures, basement membrane, and multilayered epithelium were examined using confocal fluorescence microscopy and immunohistochemistry. The potential ingrowth in vivo was simulated using the chorioallantoic membrane model (CAM assay) in ovo. ESWT on culture day 19 of oral mucosa equivalents resulted in slightly increased cellular metabolic activity. The in vitro development of basement membrane and multilayer epithelium was stimulated by ESWT. Additionally, in the CAM assay, ESWT led to a more pronounced multilayered epithelium. Thus, ESWT stimulated the formation of a more distinct and differentiated multilayered epithelium of oral mucosa equivalents in vitro and might increase the chance of efficient ingrowth, survival, and functionality of tissue equivalents in vivo.

## 1. Introduction

In reconstructive surgery, oral mucosa is used to reconstruct congenital or acquired tissue defects in various surgical fields. It is used as a graft tissue in reconstructions of the urethra [1]. Moreover, the replacement and repair of mucosa tissue in oral- and maxillofacial surgery, and also gynecology, is conceivable [2]. Nevertheless, the avalability of autologous oral mucosa tissue used for this purpose is limited and can be negatively affected by a variety of factors. In addition, tissue harvesting may cause additional injuries to the tissue-donor [3]. The in vitro generation of oral mucosa equivalents by tissue engineering using autologous cells represents a suitable alternative to transplantation of oral mucosa [4]. Oral mucosa equivalents can be generated on a collagen matrix using fibroblasts and epithelial cells of oral mucosa origin in combination with microvascular endothelial cells [5]. The structural composition of the tissue equivalents is comparable to autologous oral mucosa.

The fibroblasts enclosed in the tissue equivalent secrete cytokines and extracellular matrix and they form the necessary connective tissue required to engineer a functional mucosa tissue [6,7]. On top of the connective tissue, a multilayered epithelium must be formed by the epithelial cells. The epithelial layers show various patterns of differentiation [8]. The basal epithelial cell layer contains undifferentiated dividing cells that grow in clusters and express cytokeratin 19. Differentiating epithelial cells are located suprabasally and express cytokeratin 13. Fully differentiated epithelial cells are located on the outer layer and express involucrin [8,9,10,11]. Tight junctions are essential components of the epithelium, as they ensure the barrier function. The occurrence of tight junctions, accompanied by occludin expression in the tissue equivalent, demonstrates the functionality of the epithelium [12,13]. The epithelium acts as a barrier against external microorganisms and pathogens [14]. Epithelial cells synthesize proteins, such as collagen IV and laminin to form a basement membrane that separates the epithelium from the underlying connective tissue. Thus, they contribute to tissue organization and stability [15].

Endothelial cells are key components in tissue engineering, as they facilitate the formation of vascular-like structures, and thus the pre-vascularization of the tissue equivalents in vitro [5,16]. Vascular structures are crucial for the oxygen and nutrient supply of the engineered tissues. The pre-vascularization enables a more rapid anastomosis to the recipients vascular system and reduces the risk of necrosis after transplantation [17,18]. Thus, it is important to create tissue conditions favoring pre-vascularization and epithelium formation in vitro to enhance the success of the tissue-engineered oral mucosa equivalents when used in clinical applications. The survival of the epithelium in vivo is essential for the functionality of the tissues [16].

While extracorporeal shock wave therapy (ESWT) is successful in orthopedic applications to support healing processes, less is known about the effects of ESWT on soft tissue healing. It has been shown that ESWT exerts positive effects on angiogenesis and epithelium formation [19,20]. The in vitro application of ESWT may improve and accelerate the formation and differentiation of the multilayered epithelium of the oral mucosa equivalents. Shock waves are mechanical non-linear pressure waves that propagate in a fluid medium which reversibly deforms or changes its density [21]. In clinical applications, ESWT is an established treatment method used for various orthopedic and soft tissue conditions [22,23]. The shock waves are usually applied by placing the metal applicator of the ESWT handpiece on the affected tissue. Shock waves are generated by firing a projectile within a guiding tube that strikes the metal applicator, which in turn transmits energy after reaching the tissue boundary resulting in the stimulation of tissue regeneration [20,21,24]. The exact mechanisms behind ESWT are unknown. The use of ESWT in cell cultures and animal studies have shown that the treatment has a positive effect on in vitro cell proliferation and differentiation of various cell types [25,26,27]. After ESWT treatment in vivo, the thickening of the skin layers in rats was observed [19]. In other studies, ESWT has been shown to stimulate growth factor production and to influence signaling pathways that positively affect angiogenesis and tissue regeneration [28,29].

The purpose of the present study was to analyze the impact of ESWT on complex oral mucosa equivalents during in vitro culture with respect to the development and differentiation of a multilayered epithelium as well as on cellular survival during in vitro culture to ensure the functionality of the graft in vivo. This would offer a promising alternative for autologous mucosa tissue for reconstructive surgery.

## 2. Materials and Methods

### 2.1. Experimental Design

This study investigated the effect of ESWT during the manufacturing process of oral mucosal equivalents. For in vitro analyses, bi-cultures of endothelial cells and fibroblasts, and tri-cultures of endothelial cells, fibroblasts, and epithelial cells were prepared on a collagen matrix. These were treated with ESWT on culture day 15, culture day 19, or culture day 26. Two days after ESWT, analysis of cellular metabolic activity was performed. Untreated bi- and tri-cultures served as controls. The cell cultures on collagen matrices continued to be cultured until culture day 33. Immunofluorescence and immunohistochemical analyses of the ESWT-treated and untreated samples were then performed (Figure 1A). For in ovo analyses, bi-cultures of fibroblasts and epithelial cells and tri-cultures of fibroblasts, epithelial cells, and endothelial cells were prepared. These were treated with ESWT on culture day 19. The cell-seeded collagen matrices were cultured until culture day 33, when they were placed onto CAM and subsequently subjected to immunohistochemical analysis (Figure 1B).

### 2.2. Cells and Cell Culture

Epithelial cells were isolated from human oral mucosa by mechanical tissue processing as described previously [30]. The epithelial cells were cultivated in CnT-PRIME (CELLnTEC, Bern, Switzerland) with 100 U/100 μg/mL Penicillin-Streptomycin. 24 h before seeding onto collagen matrices, culture medium for the epithelial cells was changed to FAD_complete_ medium (60% DMEM, 30% Gibco™ DMEM/F-12 (ThermoFisher Scientific, Waltham, MA, USA), 10% Gibco™ Fetal Calf Serum (ThermoFisher Scientific, Waltham, MA, USA), 100 U/100 mg/mL Penicillin/Streptomycin, 22 mg/mL adenine (Sigma-Aldrich, St. Louis, MO, USA), 7.4 ng/mL cholera toxin (Sigma-Aldrich, St. Louis, MO, USA), 9 ng/mL EGF (Sigma-Aldrich, St. Louis, MO, USA), 36 ng/mL hydrocortisone (Sigma-Aldrich, St. Louis, MO, USA) and 5.4 µg/mL insulin (Sigma-Aldrich, St. Louis, MO, USA)).

Fibroblasts were also isolated from human oral mucosa by mechanical tissue processing. Tissue preparation for cell isolation started with the separation of the epithelium from the connective tissue of the oral mucosa. For the isolation of fibroblasts, the connective tissue was used following the same protocol for epithelial cell isolation [30]. DMEM (Sigma-Aldrich, St. Louis, MO, USA) supplemented with 10% Gibco™ Fetal Calf Serum (ThermoFisher Scientific, Waltham, MA, USA) and 100 U/100 mg/mL Penicillin/Streptomycin (Sigma-Aldrich, St. Louis, MO, USA) was used for fibroblast cultivation. At total of 24 h prior to seeding fibroblasts onto collagen matrices, the culture medium for fibroblasts was changed to endothelial cell medium (Endothelial Cell Growth Medium 2 (Promocell, Heidelberg, Germany) supplemented with 15% Gibco™ Fetal Calf Serum (ThermoFisher Scientific, Waltham, MA, USA), 100 U/100 μg/mL Penicillin-Streptomycin, 2.5 ng/mL bFGF (Sigma-Aldrich, St. Louis, MO, USA), 10 μg/mL Na-heparin (Sigma-Aldrich, St. Louis, MO, USA), and 28 mM HEPES (Sigma-Aldrich, St. Louis, MO, USA)).

Endothelial cells were isolated from human juvenile foreskin by enzymatic digestion, as described by Heller et al. [7], and were cultivated in endothelial cell medium (Endothelial Cell Growth Medium 2 (Promocell, Heidelberg, Germany) supplemented with 15% Gibco™ Fetal Calf Serum (ThermoFisher Scientific, Waltham, MA, USA), 100 U/100 μg/mL Penicillin-Streptomycin, 2.5 ng/mL bFGF (Sigma-Aldrich, St. Louis, MO, USA), 10 μg/mL Na-heparin (Sigma-Aldrich, St. Louis, MO, USA), and 28 mM HEPES (Sigma-Aldrich, St. Louis, MO, USA)).

Oral mucosa was provided by the Department of Oral- and Maxillofacial and Plastic Surgery, and the foreskin by the Department of Urology of the University Medical Center of the Johannes Gutenberg University Mainz, Germany. The primary cells were processed strictly anonymously without recording patient-related data and in accordance with the Declaration of Helsinki and approved by the local ethics committee (Landesärztekammer Rheinland-Pfalz, No. 2021-15794_1). Informed consent was obtained from each patient. Fibroblasts from oral mucosa were used up to passage 8, endothelial cells and epithelial cells up to passage 5.

### 2.3. Generation of Oral Mucosa Equivalents

For the generation of oral mucosa equivalents (tri-cultures), primary fibroblasts, endothelial cells, and epithelial cells were seeded onto bilayered collagen matrices (Bio-Gide^®^, Geistlich Pharma AG, Wolhusen, Switzerland). The collagen matrices were punched into pieces with an area of 28.3 mm^2^ and a diameter of 6 mm. They were rehydrated with serum-free medium for 20 min. On day 1, 5.6 × 10^4^ fibroblasts/matrix were seeded onto the porous side of the collagen matrix. On day 4, 1.12 × 10^5^ epithelial cells/matrix were seeded on the cell occlusive side. After 4 h, the matrices were transferred to a transwell system (TC-Inserts, 24 Well, PET 0.4 µm, TP (Sarstedt, Nümbrecht, Germany)). The cell occlusive side was directed upwards, so that the epithelial cells were at the liquid–air boundary. Epithelial cells within the insert received FAD_complete_ medium and fibroblasts on the other side of the matrices received endothelial cell medium from the plate well. On day 12, 1.12 × 10^5^ endothelial cells/matrix were seeded onto the porous side of the collagen matrix. After 4 h, the matrices were transferred back to the transwell system and positioned with the epithelial layer facing upwards. The tri-culture was incubated at 37 °C and 5% CO_2_ until culture day 33. Bi-cultures were also generated according to the same scheme, in which only two cell types were seeded on the collagen matrices, on the same culture days as described. The culture media were changed every two to three days. During cell culture, the fibroblasts and endothelial received endothelial cell medium from the plate well and the epithelial cells received FAD_complete_ medium from the insert (Figure 2).

### 2.4. Extracorporeal Shock Wave Therapy of Cells In Vitro

The cell colonized matrices were treated with extracorporeal shock waves generated from the Swiss DolorClast^®^ Classic (EMS, Nyon, Switzerland) on culture day 15, day 19, and day 26 (3, 7, and 14 days after endothelial cell seeding). For this purpose, a 1% agarose gel (agarose (Carl Roth, Karlsruhe, Germany)) in 1× Tris-boric acid-EDTA buffer (Carl Roth, Karlsruhe, Germany) was poured about 2 cm high into a 6-well plate (Greiner, Kremsmünster, Austria). An area of the size of the collagen matrix was punched out, the tissue equivalent was placed into the punched-out area and covered with an agarose gel plug. The metal applicator of the Swiss DolorClast^®^ Classic handpiece was placed directly onto the agarose gel. The tissue equivalents were treated by the shock waves through the gel with an air pressure of 3.0 bar, a frequency of 5 Hz, and an energy flux density of 0.12 mJ/mm^2^ and 500 impulses in line with the treatment procedure described by Heimes et al. [20]. Afterwards, the tissue equivalents were transferred back into the transwell system and cultured until day 33 (Figure 3). The effect of ESWT on day 15, day 19, or day 26 was examined for its effects on the cellular metabolic activity and by the effects on vascularization of the tissue equivalents. Further immunohistochemistry studies were performed after ESWT on day 19, since ESWT at this time point provided the most promising results.

### 2.5. Cellular Metabolic Activity Assay

Two days after each ESWT, the cellular metabolic activity of the oral mucosa equivalents was quantified by the AlamarBlue^TM^ cell viability assay (ThermoFisher Scientific, Waltham, MA, USA). For this purpose, the equivalents were transferred from the transwell system to a 96-well plate (Greiner, Kremsmünster, Austria), and 200 µL of AlamarBlue^TM^ (1:10 in endothelial cell medium) was added. After 4 h, the equivalents were rinsed with DPBS (ThermoFisher Scientific, Waltham, MA, USA) and transferred back into the transwell system. The fluorescence of supernatants was measured with a microplate reader at a wavelength of 590 nm (GloMax^®^ Microplate Reader, Promega, Madison, WI, USA).

### 2.6. Chick Chorioallantoic Membrane-Assay

Fertilized white Leghorn chicken eggs (LSL Rhein-Main GmbH, Dieburg, Germany) were placed horizontally into an incubator (type 400, bruja, Hammelburg, Germany) and incubated with a stable temperature of 38.5 °C and a humidity of 50–60%. On the 3rd day of incubation, 6 mL of egg white was removed from the blunt pole of the egg using a cannula (B. Braun Melsungen, Germany). Afterwards, the puncture site was tightly sealed with leukosilk^®^ (BSN medical GmbH, Hamburg, Germany). An oval window of about 2–3 cm was excised from the upper side of the egg. The window was sealed with parafilm (Sigma-Aldrich, St. Louis, MO, USA). On the 4th day of incubation, oral mucosa equivalents were placed onto the chorioallantoic membrane (CAM) with the cell occlusive side facing upwards. The equivalents remained on the CAM until embryonic day 9–14, depending on the survival time of the embryo. The tissue equivalents and the adjacent CAM were then carefully dissected. The samples were fixed with ROTI^®^Histofix 4% (Carl Roth, Karlsruhe, Germany) for 1 h and then transferred to DPBS for further analysis. The embryos were decapitated according to ethical protocols before disposal.

### 2.7. Immunofluorescence

For immunofluorescence analyses, the fixed oral mucosa equivalents were incubated in 0.2% Triton-X (Sigma-Aldrich, St. Louis, MO, USA) at room temperature (RT) for 20 min. Afterwards, the tissue equivalents were washed with DPBS three times for 10 min at 300 rpm on a thermoshaker (Cellmedia, Carl Roth, Karlsruhe, Germany). This was followed by incubation with CD31 antibody (monoclonal mouse anti-human, clone JC70A, Dako, Santa Clara, CA, USA, 1:50 in 1% Bovine Serum Albumin (BSA) solution (Sigma-Aldrich, St. Louis, MO, USA) for 1 h at RT and afterwards overnight at 4 °C on the thermoshaker at 300 rpm. The next day, the tissue equivalents were washed with DPBS three times for 10 min and incubated with the secondary antibody Alexa Fluor^®^ 488 (polyclonal goat anti-mouse, ThermoFisher Scientific, Waltham, MA, USA, 1:200 in 1% BSA solution) for 2 h at RT on the thermoshaker at 300 rpm. The tissue equivalents were then washed three times with DPBS, and vascular-like structures were analyzed with confocal fluorescence microscopy (TCS SP8, (Leica Microsystems, Wetzlar, Germany)).

### 2.8. Immunohistochemistry

The fixed oral mucosa equivalents were embedded in paraffin for immunohistochemically analyses and cut transversely to the porous and occlusive side of the matrices (section thickness 4 µm). Consecutive sections from the samples were used for the staining with different antibodies. Paraffin was removed and samples were hydrated with a descending alcoholic series. This was followed by heat-inducing demasking with 1× Target Retrieval Solution, pH 9 in deionized water (Agilent Technologies, Santa Clara, CA, USA). Next, endogenous peroxidase was blocked with 3% H_2_O_2_ (Carl Roth, Karlsruhe, Germany) solution in deionized water for 10 min at RT. Afterwards, the samples were washed twice with deionized water and once with DPBS. After this, non-specific antibody binding was blocked with 1% BSA solution in DPBS and the samples were incubated in a humidity chamber at RT for 1 h. Thereafter, the samples were rinsed twice with deionized water, followed by incubation with the respective primary antibody: monoclonal mouse anti-human CD31 for endothelial cell staining (clone JC70A, Dako, Santa Clara, CA, USA, 1:50), monoclonal mouse anti-collagen IV as a marker for basement membrane (clone COL-94, abcam, Cambridge, UK, 1:250), monoclonal mouse anti-laminin as a marker for basement membrane (clone LAM-89, Sigma-Aldrich, St. Louis, MO, USA, 1:1000), polyclonal rabbit anti-occludin as a marker for tight junctions (abcam, Cambridge, UK, 1:300), monoclonal mouse anti-cytokeratin 19 as a marker for undifferentiated epithelial cells (clone RCK108, Dako, Santa Clara, CA, USA, 1:100), monoclonal rabbit anti-cytokeratin 13 as a marker for suprabasally differentiating epithelial cells (clone EPR3671, abcam, Cambridge, UK, 1:500), monoclonal mouse anti-involucrin as a marker for fully differentiated epithelial cells (clone SY5, ThermoFisher Scientific, Waltham, MA, USA, 1:200). The antibodies were diluted with Dako REALTM Antibody Diluent (Dako, Santa Clara, CA, USA) and incubated in the humidity chamber overnight at 4 °C. The samples were washed three times with DPBS for 2 min, incubated with Dako REAL EnVision HRP RABBIT/MOUSE (Dako, Santa Clara, CA, USA) in the humidity chamber for 30 min at RT and again washed three times for 2 min with DBPS. The samples were then incubated with DAB solution (DAB+CHROMOGEN, 1:50 in Dako REAL SUBSTRATE BUFFER (Dako, Santa Clara, CA, USA)) for a maximum of 5 min until staining was evident. Subsequently, the sections were washed three times for 2 min with deionized water, followed by cell nuclear staining with Mayer’s hemalaun (Carl Roth, Santa Clara, CA, USA) for 10 min. Afterwards the sections were held under running water for 10 min and rinsed with deionized water, followed by an ascending alcohol series. Finally, the samples were covered with Hico-Mic (Hico, Köln, Germany) and assessed by digital light microscopy (VHX-1000D, (Keyence, Osaka, Japan).

### 2.9. Statistical Analysis

Statistical analyses were performed with Microsoft Excel (Excel 2016, Microsoft Corporation, Redmond, WA, USA). Significant differences were calculated using the two-sided Student *t*-test and represented as mean values ± standard deviation of the mean (SD).

## 3. Results

### 3.1. Cellular Metabolic Activity after Extracorporeal Shock Wave Therapy

When cell colonized matrices were treated with ESWT on culture day 15, the cellular metabolic activity of the bi-culture of fibroblasts and endothelial cells was non-significantly decreased to 96% (±23%) and that of the tri-culture of fibroblasts, epithelial cells, and endothelial cells was non-significantly decreased to 86% (±7%) compared to the respective untreated control. When bi- and tri-cultures were treated with ESWT on culture day 19, the cellular metabolic activity of the bi-culture was non-significantly increased to 106% (±11%) and that of the tri-culture was non-significantly increased to 109% (±4%). When bi- and tri-cultures were treated with ESWT on culture day 26, the cellular metabolic activity of the bi-culture non-significantly increased to 105% (±28%) and that of the tri-culture was non-significantly increased to 124% (±39%) (Figure 4).

### 3.2. Vascular-like Structures after Extracorporeal Shock Wave Therapy

To determine the best time for application of ESWT with respect to vessel formation, we evaluated the influence of ESWT on the endothelial cell response. Treatment of bi-cultures of fibroblasts and endothelial cells with ESWT on culture day 15 resulted in a marked deterioration of vascular-like structures on culture day 33 compared to the untreated control. Additionally, the treatment of bi-cultures with ESWT on culture day 26 led to a deterioration of the vascular-like structure formation on culture day 33 (Figure 5A). Treatment of bi-cultures with ESWT on culture day 19 led to similar pronounced vascular-like structures on culture day 33 as those found in the untreated control culture (Figure 5A). Treatment of tri-cultures, which contained fibroblasts, epithelial cells, and endothelial cells (oral mucosa equivalent) with ESWT led to less formation of vascular-like structures on culture day 33 than in bi-cultures at all three treatment time points (Figure 5B). Furthermore, treatment of tri-culture with ESWT, led to no differences in the depth of localization of the endothelial cells on culture day 33 compared to the untreated control (Figure 5C). After treatment with ESWT of tri-cultures on culture day 19, at least fragments of the vascular-like structures were still present on culture day 33 (Figure 5B). Based on these results, oral mucosa equivalents were subsequently treated with ESWT on culture day 19 (Figure 1).

### 3.3. Formation of Multilayered Epithelium after Extracorporeal Shock Wave Therapy on Oral Mucosa Equivalent In Vitro

To assess the formation of a multilayered epithelium after treatment of the oral mucosa equivalents with ESWT during in vitro culture, cytokeratin 19 was used as a marker for undifferentiated epithelial cells, cytokeratin 13 served as a marker for suprabasally differentiating epithelial cells, and involucrin expression was a marker of fully differentiated epithelial cells. Treatment of oral mucosa equivalents (tri-culture) with ESWT on culture day 19 led to a more pronounced cytokeratin 19 staining on culture day 33 compared to the untreated control. Cytokeratin 13 and involucrin expression were also stronger after treatment with ESWT in comparison to the untreated control. Clearly recognizable was the co-localization of cytokeratin 19, cytokeratin 13, and involucrin in the oral mucosa equivalents (Figure 6).

### 3.4. Formation of Basement Membrane after Extracorporeal Shock Wave Therapy on Oral Mucosa Equivalent In Vitro

To assess the formation of a basement membrane and the functionality of the epithelium after treatment with ESWT on oral mucosal equivalents during in vitro culture, collagen IV and laminin were used as markers for formation of the basement membrane. Occludin is a key component in tight junctions and was used as marker to analyze the functionality of the epithelial cell layers. Treatment of oral mucosa equivalents (tri-cultures) with ESWT on culture day 19 resulted in a more distinct expression of collagen IV on culture day 33 compared to the untreated control. The expression of laminin was slightly stronger after ESWT compared to the untreated control. The treatment with ESWT exerted a positive effect on the occludin expression in comparison to untreated control (Figure 7).

### 3.5. Impact of Extracorporeal Shock Wave Therapy on Multilayered Epithelium in Oral Mucosa Equivalents In Ovo in the Chorioallantoic Membrane

Cell colonized collagen matrices with bi-cultures of fibroblasts and epithelial cells, or tri-cultures (oral mucosa equivalents) of fibroblasts, epithelial cells, and endothelial cells (EC) were transferred to the CAM on culture day 33. When the bi-cultures were treated with ESWT on culture day 19, we observed an increased cytokeratin 19 and cytokeratin 13 expression after in ovo culture. Untreated bi-cultures showed a weak cytokeratin 19 and cytokeratin 13 expression during in ovo culture. In contrast, the treatment of tri-cultures with ESWT led to a poor expression of cytokeratin 19 and cytokeratin 13 layers during in ovo culture, compared to untreated tri-cultures after in ovo cultivation. The co-localization of cytokeratin 19 and cytokeratin 13 was clearly detected in all samples (Figure 8). Due to the treatment with ESWT, a slightly more pronounced expression of involucrin in the bi-cultures was detected compared to untreated bi-cultures. In contrast, the treatment with ESWT of tri-cultures led to a lower expression of involucrin after in ovo cultivation, compared to the untreated tri-cultures (Figure 8). The expression of the epithelial cell markers cytokeratin 19 and cytokeratin 13 was more pronounced in untreated tri-cultures than in untreated bi-cultures after in ovo culture on CAM. The expression of involucrin was equally expressed in the untreated bi-cultures as in the untreated tri-culture (Figure 8). The presence of endothelial cells in the tri-cultures offered a clear advantage with respect to the expression of the generation of a differentiated, multilayered epithelium during in ovo culture.

### 3.6. Impact of Extracorporeal Shock Wave Therapy on Basement Membrane in Oral Mucosa Equivalents In Ovo in CAM Model

Cell colonized collagen matrices with bi-cultures of fibroblasts and epithelial cells, or tri-cultures (oral mucosa equivalents) of fibroblasts, epithelial cells, and endothelial cells (EC) were transferred to the CAM on culture day 33. The treatment with ESWT on culture day 19 led to a slight collagen IV and laminin staining in the bi-cultures after the transplantation on the CAM. In untreated bi-cultures, no collagen IV or laminin layer was detected. In contrast, in ESWT-treated tri-cultures, no collagen IV or laminin layer was detected after in ovo cultivation. In untreated tri-culture, collagen IV and laminin expression were detected during in ovo culture (Figure 9). In ESWT-treated bi-cultures, a more adequate occludin pattern on the CAM was detected, than in untreated bi-cultures. In untreated bi-cultures, occludin was detectable in a dispersible pattern after transfer on CAM for in ovo culture. In tri-cultures, treatment with ESWT led also to an adequate occludin expression in ovo (Figure 9). Expression of collagen IV, laminin, and involucrin was more pronounced in untreated tri-cultures than in untreated bi-cultures after in ovo culture on CAM (Figure 9). The presence of endothelial cells in the tri-cultures offered a clear advantage with respect to the expression of the formation a basement membrane during in ovo culture.

### 3.7. Impact of Extracorporeal Shock Wave Therapy on Endothelial Cells in Oral Mucosa Equivalents In Ovo in the CAM Model

The treatment of oral mucosa equivalents with ESWT during in vitro culture on day 19 resulted in weaker endothelial cell detection, determined by CD31 staining, after transfer on CAM for in ovo culture than in the untreated control. In untreated oral mucosa equivalents, the endothelial cells were distributed contiguously in the superficial area of the tissue equivalent. After treatment of the oral mucosa equivalents with ESWT during in vitro culture on culture day 19, a smaller number of endothelial cells were distributed in the tissue equivalents. Nevertheless, they were still present and thus survived during in ovo culture (Figure 10).

## 4. Discussion

The survival of the multilayered epithelium of tissue engineered oral mucosa equivalents in vivo is essential for the functionality of the tissues after transplantation [16]. During the cultivation process of the mucosa equivalents, it is important to provide conditions that support the epithelium formation and differentiation. The better the epithelium is developed in vitro, the greater the chances are of successful tissue transplantation. In this study, oral mucosa equivalents were generated, were treated with ESWT in vitro and evaluated for their functionality in ovo. Through the application of ESWT in vitro, we achieved a major improvement in the formation of the multilayered epithelium of the oral mucosa equivalents in vitro and in the CAM model in ovo.

Preliminary tests showed that ESWT on day 19 was suitable for the functional preservation of endothelial cells. Therefore, this time point was adopted for further experiments. The treatment with ESWT of oral mucosa equivalents on culture day 19 resulted in significantly increased expression of the epithelial markers cytokeratin 19 and cytokeratin 13, as well as an increased expression of involucrin during in vitro culture compared to the untreated mucosa equivalents. Cytokeratin 19 is expressed by the basal undifferentiated and dividing epithelial cells [8,10]. Cytokeratin 13 is expressed by suprabasally differentiating epithelial cells and involucrin by fully differentiated epithelial cells [9,11]. The fact that all three markers were increased after treatment with ESWT indicating an induction of a physiologically highly active epithelial layer. These results were consistent with observations from other studies. For example, Ottomann et al. reported that ESWT significantly accelerated epithelialization on human split thickness graft donor sides [31]. The treatment with ESWT may stimulate proliferation and differentiation of epithelial cells of oral mucosa equivalents during in vitro culture. ESWT induces several signaling pathways in cell cultures, including ERK- [32], Wnt- [33], and FAK-pathways [34]. These signaling pathways are important during wound healing processes [35,36] and may be responsible for the enhanced formation of the multilayer epithelium of the oral mucosa equivalents in vitro. In addition, shock waves induce the inhibition of glycogen synthase kinase-3β, which plays an important role in epithelial wound healing by activating the Wnt/β-catenin signaling pathway [28,29]. The treatment with ESWT led to a significantly increased expression of cytokeratin 19, cytokeratin 13 and of involucrin not only during in vitro culture, but also after transfer on CAM for in ovo cultivation. Thus, ESWT treatment during in vitro cultivation improved the formation and differentiation of a multilayered epithelium and led to a better differentiated epithelium during in ovo cultivation. Thus, ESWT could promote the survival and functionality of the tissue engineered oral mucosa equivalents after transplantation.

A higher expression of collagen IV and a slightly higher expression of laminin after transfer onto the CAM was observed for in vitro oral mucosa equivalents cultures treated with ESWT compared to the untreated equivalents. Collagen IV and laminin are important components of the basement membrane [15]. The in vitro expression of occludin, a component of tight junctions, was also more pronounced after ESWT in comparison to the untreated controls. The expression of protein components of tight junctions indicates a functional epithelial cell layer [12,13]. We observed a weak expression of collagen IV, laminin, and occludin in the untreated controls, whereas the basement membrane in healthy autologous oral mucosa tissue is characterized by a prominent expression of collagen IV and laminin [37]. Thus, the problem of the weak formation of a basement membrane in the mucosa equivalents can possibly be overcome with ESWT. This effect may also be triggered by growth factors, as their secretion is promoted by the Wnt/β-catenin signaling pathway, which, in turn, is activated by ESWT [38]. The formation of basement membrane after treatment with ESWT during in vitro culture improved its structural formation and survival during in ovo growth after transplantation on the CAM to a small extent.

We could show that the timing of application of ESWT of tissue equivalents during in vitro cell culture was of great importance with respect to its impact on epithelium formation and vascularization. The pre-vascularization of oral mucosal tissues is necessary to achieve faster and better attachment of the transplanted tissue to the host vasculatore [5,39]. We confirmed that, without ESWT, pre-vascularization of oral mucosa equivalents offered a clear advantage with respect to the expression of the basement membrane and the generation of a differentiated, multilayered epithelium during in ovo culture. After treatment with ESWT, pre-vascularization did not result in this support. It is crucial that treatment with ESWT does not disturb the growth and development of endothelial cells, to ensure the pre-vascularization of the engineered tissue products.

The total cellular metabolic activity of oral mucosa constructs increased slightly after ESWT on culture day 19 and culture day 26 compared to the untreated control. In contrast, the treatment with ESWT on culture day 15 resulted in a slightly decreased cellular metabolic activity of the mucosa equivalents. This suggests that tissue equivalents at early stages in the generation process may be more susceptible to damage by ESWT. To assess the effect of ESWT on epithelial cells within the oral mucosa equivalents, we compared the cellular metabolic activity of bi-cultures of fibroblasts and endothelial cells with the metabolic activity of tri-cultures (oral mucosa equivalents) of fibroblasts, endothelial cells, and epithelial cells treated with ESWT. The cellular metabolic activity of the tri-culture after treatment with ESWT on culture day 15 was reduced compared to those of the bi-culture. The treatment of oral mucosa equivalents with ESWT appeared to have a negative effect on cellular metabolic activity within the oral mucosa equivalents, especially on the viability of epithelial cells during early stages of the generation period of the oral mucosa equivalent. This effect may be a result of the small cell number of cells and the lower stages of differentiation of the epithelial layer during the early stage of development of the engineered oral mucosa, leading to an increased susceptibility of epithelial cells to mechanical stress [8,40]. At advanced stages of in vitro tissue formation, the cell density within the oral mucosa equivalents was higher due to proliferation, resulting in a thicker epithelial layer. This may have protected the cells from mechanical stress [8,40]. In addition, at this stage of advanced development of the engineered oral mucosa equivalents, treatment with ESWT may cause the cells to release growth factors, for example vascular endothelial growth factor (VEGF),and upregulate Wnt/β-catenin signaling pathway that had a positive effect on cellular metabolic activity and proliferation of epithelial cells [28,33]. Previous studies have shown an increased cellular metabolic activity of cell cultures after ESWT when cells were grown as monocultures, in cell culture plates, in flasks or in suspension [26,41,42,43]. In contrast, we assessed the metabolic activity of cells in a complex tissue-like 3D surrounding. Others have shown negative effects of extracorporeal shock waves on cell viability. Mattyasovszky et al. observed a lower viability of human skeletal muscle cells after the application of high energy flux densities of >0.14 mJ/mm^2^ [44], and ESWT with even higher intensity was shown to be cytoreductive [45]. The cellular metabolic activity of the tissue equivalents treated with ESWT at later stages during the generation process (on culture day 19 and day 26) was more increased in tri-culture than in bi-culture. This suggests that treatment with ESWT positively affected the cellular metabolic activity, especially of epithelial cells of the oral mucosa equivalents during in vitro culture.

Endothelial cells were affected by a too early time point of treatment with ESWT. The formation of vascular-like structures by endothelial cells in bi-cultures with fibroblasts was severely disturbed by ESWT on culture day 15 when endothelial cells were 3 days cultured on collagen matrices. This was not sufficient time to form vascular-like structures [7]. Presumably, the treatment with ESWT exposed the endothelial cells to mechanical stress and, as result, they lost the capacity to form vascular-like structures. In addition, we observed a negative impact of ESWT after 14 days of endothelial cell culture (ESWT on culture day 26). At this time point, the endothelial cells had already formed a network of vascular-like structures [7]. The vascular-like structures appeared to be negatively affected by ESWT most likely due to mechanical stress. In contrast, the treatment with ESWT on culture day 19, when endothelial cells had been cultured on the collagen matrix for 7 days and vascular-like structures may have begun to form, exhibited no negative effect from the treatment on vessel formation compared to the untreated control. Heller et al. have shown that VEGF secretion by fibroblasts in the mucosa equivalents was highest on culture day 19 and decreased thereafter. Increased VEGF concentration promotes microvascular endothelial cell viability, proliferation, migration, and vessel formation [7]. These effects may create favorable conditions for ESWT treatment of the endothelial cells on culture day 19 compared to treatment on day 15 or day 26 and may protect the cells from an adverse impact of mechanical stress due to ESWT. It is possible that the settings of the ESWT need to be adjusted for in vitro experiments. Sung et al. described a positive effect of ESWT on angiogenesis by applying 0.14 mJ/mm^2^ and 140 impulses on macrovascular endothelial cells seeded on Matrigel [46]. Nevertheless, we confirmed that the timing of the treatment with ESWT of endothelial cells to support the pre-vascularization of the oral mucosa equivalent was a crucial element during cell culture. Further studies need to be done to determine optimal conditions of ESWT for vessel formation in oral mucosa equivalents.

We observed a negative interaction between endothelial cells and epithelial cells in oral mucosa equivalents with the ESWT settings and applications used in this study. Vascular-like structures in tri-cultures of fibroblasts, endothelial cells, and epithelial cells after ESWT were significantly less pronounced than in bi-cultures of fibroblasts and endothelial cells. The presence of epithelial cells in the tissue equivalents appeared to have a negative influence on the vessel formation after ESWT. At the same time, the stimulating effect of ESWT on epithelium and basement membrane formation was disturbed by the presence of endothelial cells in the tissue equivalents during in ovo culture. The combination of pre-vascularization of oral mucosa equivalents and the treatment with ESWT disrupted the formation of a multilayered epithelium and a basement membrane in ovo. The settings and application used for the ESWT in the present study may alter the secretion of growth factors of the various cell types in the oral mucosa equivalents that negatively interfered with the proliferation and differentiation of epithelial cells in ovo and disrupted endothelial cell development. In contrast, a previous study showed that ischemic skin flaps seeded with endothelial progenitor cells and treated with ESWT promoted skin flap survival during in vivo experiments in rats [47].

In conclusion, we have shown that ESWT led to an improved formation and differentiation of a multilayered epithelium in vitro and a better differentiated epithelium during cultivation in ovo. Further experiments are needed to identify the optimal settings of ESWT (ideal energy flux densities, frequencies, and impulses) for tissue engineered pre-vascularized oral mucosa equivalents. In addition, studies are needed to determine the underlying mechanisms of the interactions between epithelial and endothelial cells after treatment with ESWT. Based on the promising results of the present study, animal studies are necessary to validate the model system prior to being applied in humans.

## 5. Conclusions

The in vitro application of ESWT during tissue engineering of oral mucosa equivalents promoted the formation and differentiation of a multilayered epithelium and a basement membrane during in vitro cultivation. ESWT treatment resulted in enhancement of the formation of a basement membrane and especially for the differentiation of the multilayered epithelium in the CAM model during in ovo culture. The metabolic activity of the cells within the tissue equivalents was stimulated. The optimal timing of the ESWT during the tissue engineering process was crucial. Our results have the potential to revolutionize the field of tissue engineering, as the application of ESWT during in vitro culture may stimulate the differentiation and vascularization of tissue equivalents and, thus, may promote graft acceptance and graft functionality after transplantation into the recipient.

## Figures and Tables

**Figure 1 biomedicines-10-00700-f001:**
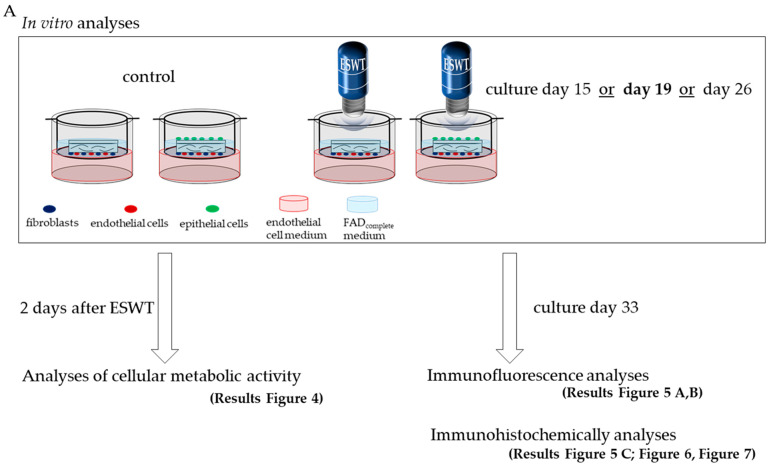
Schematic illustration of the experimental design. (**A**) In vitro analyses were performed to investigate the influence of ESWT during the manufacturing process of the oral mucosa equivalent. (**B**) In ovo analyses were performed to investigate the impact of ESWT on oral mucosa equivalents for in vivo applications.

**Figure 2 biomedicines-10-00700-f002:**
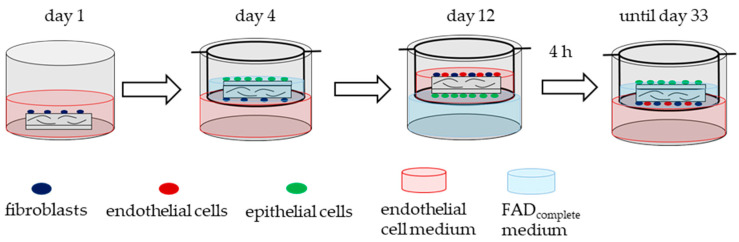
Schematic illustration of the generation of oral mucosa equivalents (tri-culture). Fibroblasts, epithelial cells, and endothelial cells were seeded at different days on a collagen matrix and cultured in a transwell system with two different cell culture media for 33 days.

**Figure 3 biomedicines-10-00700-f003:**
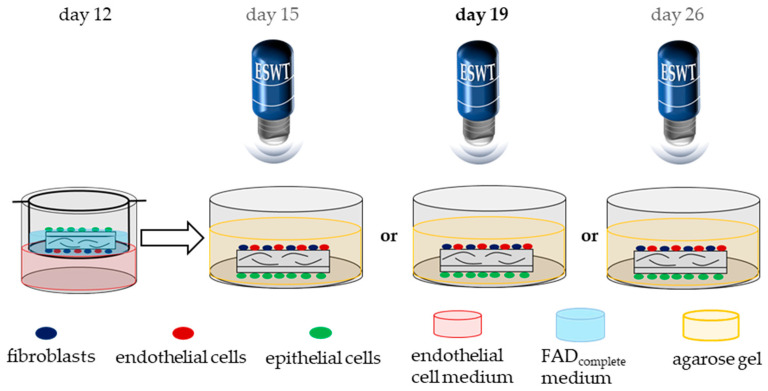
Schematic illustration of the method of application of ESWT to the cell colonized matrices. The cell-colonized matrices were treated with ESWT on day 15, day 19, or day 26 of cell cultivation. After treatment, the tissue equivalents were transferred back into the transwell system until culture day 33. The treatment on day 19 provided the most promising results. Thus, further evaluation of the impact of ESWT on the epithelium and the basement membrane were assessed with the day 19 treatment scheme.

**Figure 4 biomedicines-10-00700-f004:**
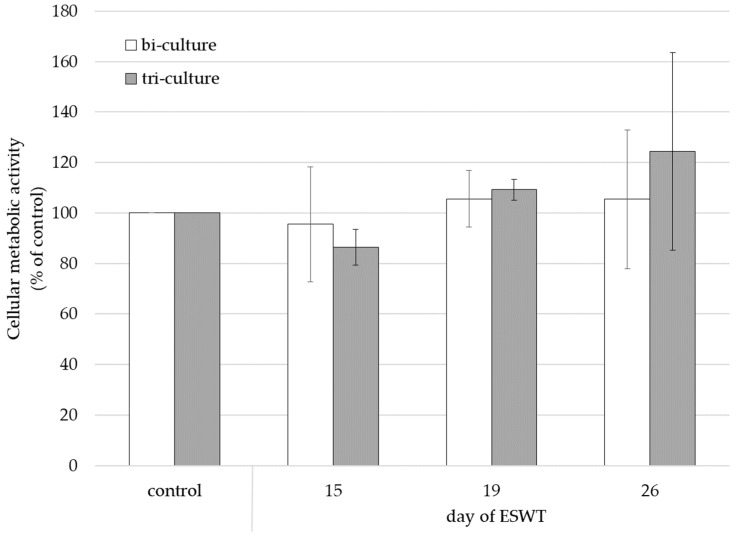
Cellular metabolic activity of bi-cultures of endothelial cells and fibroblasts or tri-cultures of endothelial cells, fibroblasts, and epithelial cells on collagen matrices two days after ESWT on day 15, day 19, or day 26 of cultivation. Untreated bi- and tri-cultures served as control. Cellular metabolic activity was examined by AlamarBlue^TM^ cell viability assay. Data are expressed as means ± SD. Measurements were conducted in duplicate and repeated three times.

**Figure 5 biomedicines-10-00700-f005:**
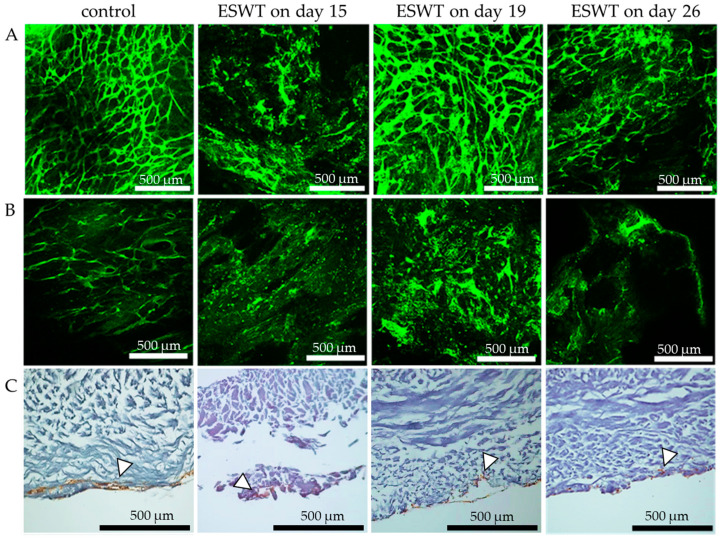
Formation of vascular-like structures and localization of endothelial cells on day 33 of bi-culture and tri-culture on collagen matrices after treatment with ESWT. (**A**) Bi-cultures of fibroblasts and endothelial cells or (**B**) Tri-cultures of fibroblasts, epithelial cells and endothelial on collagen matrices were treated with ESWT on culture day 15, day 19, or day 26 and fixed on culture day 33. Untreated bi- or tri-culture fixed on culture day 33 served as the control. Vascular-like structures were analyzed with confocal fluorescence microscopy after CD31 staining. (**C**) The localization of endothelial cells in untreated and ESWT treated tri-cultures were analyzed by immunohistochemical CD31 staining (white arrows). In the images, the cell occlusive side of the collagen matrix is directed upwards, and the porous side is directed downwards. Scale bars represent 500 µm.

**Figure 6 biomedicines-10-00700-f006:**
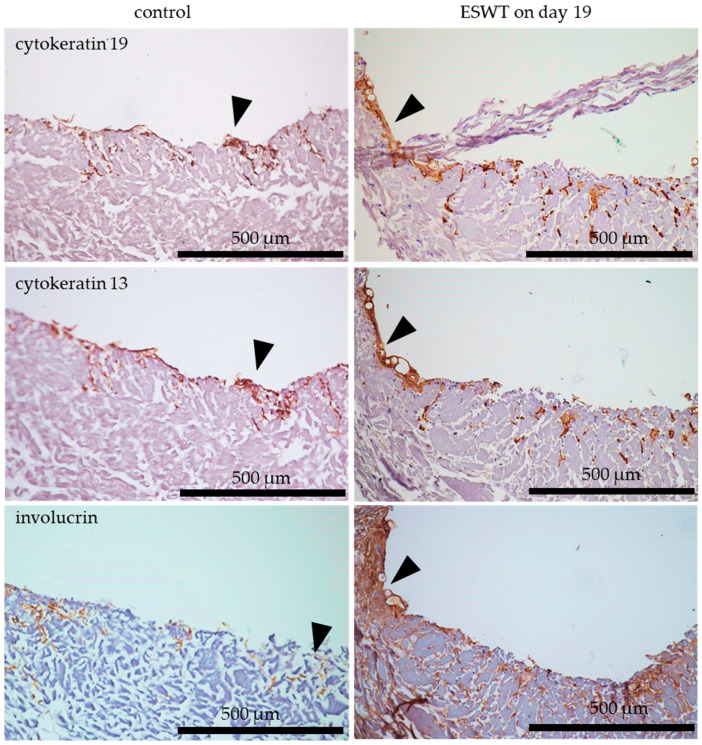
Formation of a multilayered epithelium on culture day 33 of oral mucosa equivalent (tri-culture) in vitro after treatment with ESWT on culture day 19. Untreated tissue equivalents fixed on day 33 served as control. Cytokeratin 19 was used as a marker for undifferentiated basal epithelial cells (arrows), cytokeratin 13 for suprabasally differentiating epithelial cells (arrows), and involucrin for fully differentiated epithelial cells (arrows). The oral mucosa equivalents were analyzed by immunohistochemistry. Consecutive sections from the samples were used for the staining with different antibodies. In the images, the cell occlusive side of the collagen matrix is directed upwards, and the porous side is directed downwards. Scale bars represent 500 µm.

**Figure 7 biomedicines-10-00700-f007:**
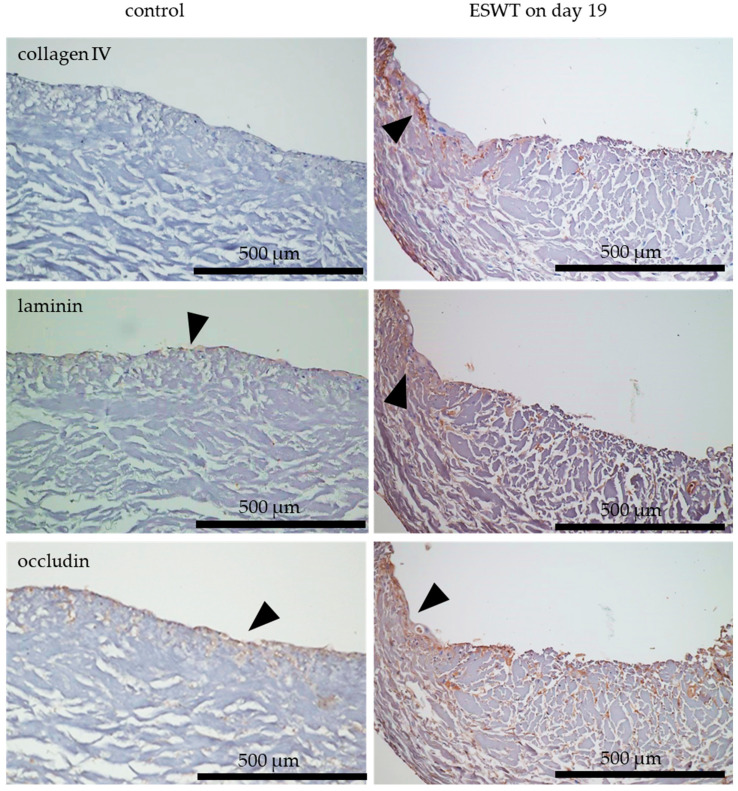
Formation of a basement membrane in oral mucosa equivalents (tri-culture) on culture day 33 in vitro after ESWT on culture day 19. Untreated tissue equivalents fixed on day 33 served as control. Collagen IV (arrows) and laminin (arrows) were used as makers of the basement membrane. Occludin (arrows) is present in tight junctions and was used to analyze the functionality of the epithelial cell layers. The oral mucosa equivalents were analyzed by immunohistochemistry. Consecutive sections from the samples were used for the staining with different antibodies. In the images, the cell occlusive side of the collagen matrix is directed upwards, and the porous side is directed downwards. Scale bars represent 500 µm.

**Figure 8 biomedicines-10-00700-f008:**
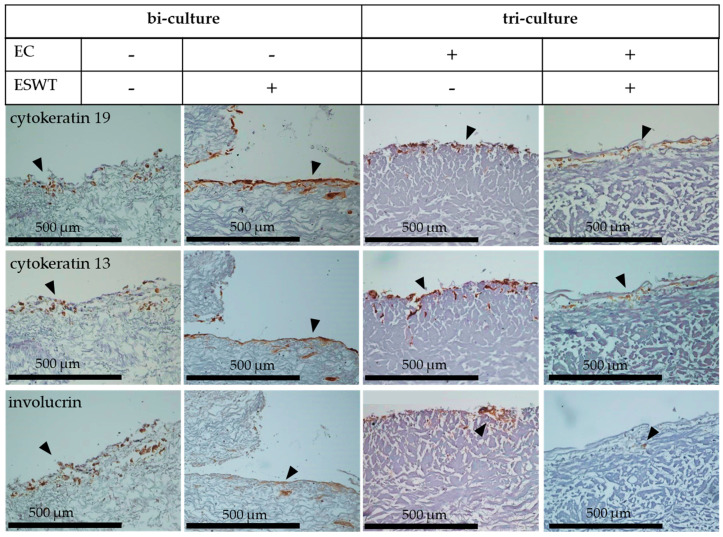
Formation of a multilayered epithelium in bi-cultures (fibroblasts and epithelial cells) and in oral mucosa equivalents (tri-cultures of fibroblasts, epithelial cells, and endothelial cells) in ovo after treatment with ESWT on culture day 19. The formation of the multilayered epithelium of tissue equivalents on the chorioallantoic membrane (CAM) was examined in ESWT treated and untreated bi-cultures of fibroblasts and epithelial cells, and in ESWT treated and untreated oral mucosa equivalents (tri-cultures) of fibroblasts, epithelial cells, and endothelial cells (EC). Cytokeratin 19 was used as a marker for undifferentiated basal epithelial cells (arrows), cytokeratin 13 for suprabasally differentiating epithelial cells (arrows), and involucrin for fully differentiated epithelial cells (arrows). Consecutive sections from the samples were used for the staining with different antibodies. In the images, the cell occlusive side of the collagen matrix is directed upwards, and the porous side directed downwards. The bi-cultures and oral mucosa equivalents were analyzed by immunohistochemistry. Scale bars represent 500 µm.

**Figure 9 biomedicines-10-00700-f009:**
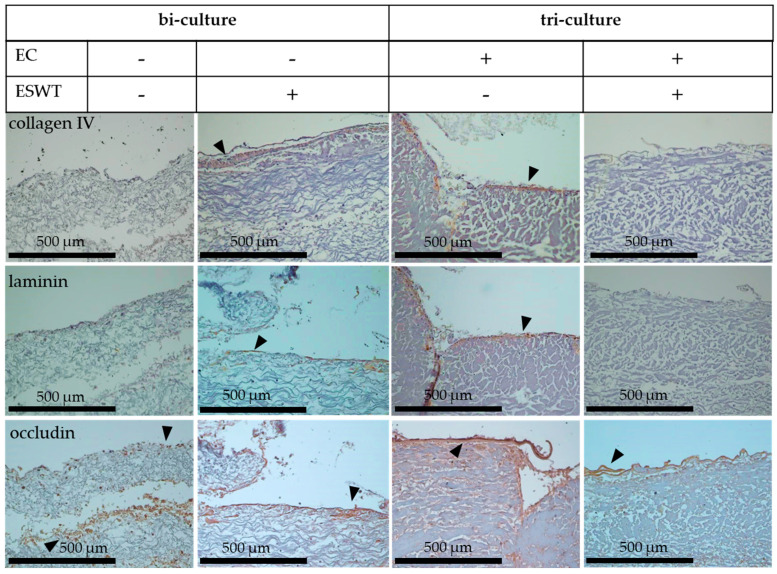
Formation of a basement membrane in bi-cultures (fibroblasts and epithelial cells) and in oral mucosa equivalents (tri-cultures of fibroblasts, epithelial cells, and endothelial cells) in ovo after treatment with ESWT on culture day 19. The formation of the basement membrane of tissue equivalents on chorioallantoic membrane (CAM) was examined in ESWT treated and untreated bi-cultures of fibroblasts and epithelial cells, and in ESWT treated and untreated oral mucosa equivalents (tri-cultures) of fibroblasts, epithelial cells, and endothelial cells (EC). Collagen IV (arrows) and laminin (arrows) were used as markers of the basement membrane. Occludin (arrows) is present in tight junctions and was used to analyze the functionality of the epithelial cell layers. The bi-cultures and oral mucosa equivalents were analyzed by immunohistochemistry. Consecutive sections from the samples were used for the staining with different antibodies. In the images, the cell occlusive side of the collagen matrix is directed upwards, and the porous side directed downwards. Scale bars represent 500 µm.

**Figure 10 biomedicines-10-00700-f010:**
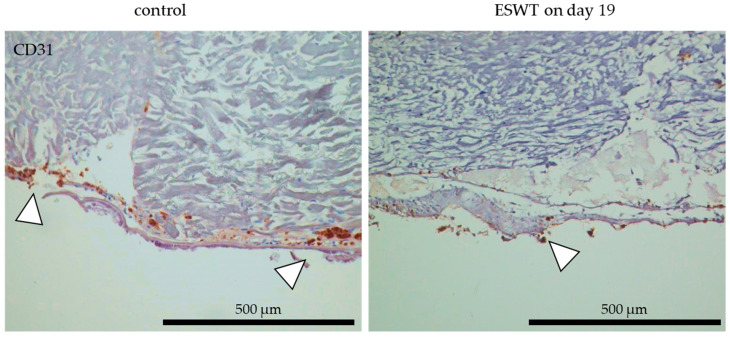
Localization of human microvascular endothelial cells in oral mucosa equivalents in ovo after ESWT in vitro. Oral mucosa equivalents were ESWT-treated on in vitro culture day 19 and transferred to the chorioallantoic membrane (CAM) on culture day 33. Untreated tissue equivalents on CAM served as the control. Endothelial cells were examined by CD31 staining (white arrows) with immunohistochemistry. In the images, the cell occlusive side of the collagen matrix is directed upwards and porous side directed downwards. Scale bars represent 500 µm.

## Data Availability

All data can be requested from the corresponding author.

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
