# Peer review of "Extracorporeal Shock Wave Therapy Improves In Vitro Formation of Multilayered Epithelium of Oral Mucosa Equivalents"

_biomedicines, 2022, doi:10.3390/biomedicines10030700_

Round 1
Reviewer 1 Report
The Authors have submitted a manuscript that describe an exogenous shock wave approach for the composition of oral mucosa equivalent multilayered epithelium.
The manuscript is interesting, and the topic addressed is of relevance for biomedical research. The employment of alternative in vivo models is of special merit as in agreement with the last EU suggestions. The data shown in the manuscript provide a clear understanding of the efficacy of ESWT therapy in inducing epithelium formation and differentiation in vitro and its potential in vivo. However, some concerns should be addressed before the acceptance of the manuscript:
- The manuscript mostly relies on qualitative analysis. The Authors may add some quantitative analysis.
- The authors should provide a graph of embryos survival from the day of mucosa equivalents transplantation up to tissue harvesting day, as the survival of chick embryos is an indication of tissue toxicity and/or rejection (see for example, doi: 10.1021/acsptsci.1c00083).
- Figure 8 could be complemented with an in ovo pictures of the bi-culture and tri-culture prior to tissue collection and staining, as it would be useful to the readers to better understand the organization of the in vitro engineered tissue.
- Is there any advantage in using the CAM model as an in vivo system for inducing oral mucosa equivalents in growth and functionality compared to other in vivo approaches? If any, the authors should add this information in the Discussion section.
- there are some typos along the main text (for example, line: 47).
Reviewer 2 Report
This article investigates the effects of extracorporeal shock wave therapy in the field of tissue regeneration through an in vitro study. The authors focus on tissue engineering of oral mucosa, using epithelial, endothelial and fibroblast models. This topic has been investigated recently in the field of regeneration medicine. The introduction provides the proper amount of information, and the methods are described in detail. Although the authors presented many data, the data are not described clearly to guide the reader through the aim of the study. However, a thorough revision according to the following indications is requested:
- To clear the several phases of the study, an experimental design is required. Please, introduce a specific paragraph in the methods section.
- In the figures 4-5-6-7-8-9 the scale bars are reported, but their value appears only in the caption. I recommend introducing the value 500µm in each figure.
- The discussion is too long. I suggest shortening it.
- The bibliography should be revised according to the criteria of the journal (i.e. 13). Furthermore, I suggest reducing the number of references. Some of them are not recent.
- I suggest checking the English language.
Round 2
Reviewer 2 Report
The authors revised the manuscript according to the indications of the reviewer. However, I suggest moving the "Experimental design" at the beginning of the methods. Furthermore, in some figures the value of scale bar is missing. I reccomand introdusing it in each figure.